# Biomass-Derived Carbon Molecular Sieves Applied to an Enhanced Carbon Capture and Storage Process (e-CCS) for Flue Gas Streams in Shallow Reservoirs

**DOI:** 10.3390/nano10050980

**Published:** 2020-05-20

**Authors:** Elizabeth Rodriguez Acevedo, Camilo A. Franco, Francisco Carrasco-Marín, Agustín F. Pérez-Cadenas, Farid B. Cortés

**Affiliations:** 1Grupo de Investigación en Fenómenos de Superficie–Michael Polanyi, Facultad de Minas, Universidad Nacional de Colombia-Sede Medellín, Medellín 050034, Colombia; caafrancoar@unal.edu.co; 2Grupo de Investigación en Materiales Avanzados y Energía-MATyER, Facultad de Ingeniería, Instituto Tecnológico Metropolitano-ITM, Medellín 050034, Colombia; 3Research Group in Carbon Materials, Faculty of Sciences, University of Granada, 18071 Granada, Spain; fmarin@ugr.es (F.C.-M.); afperez@ugr.es (A.F.P.-C.)

**Keywords:** Adsorption, carbon dioxide (CO_2_), carbon nanospheres, enhanced carbon capture and storage (e-CCS), flue gas, molecular nano-sieves

## Abstract

It is possible to take advantage of shallow reservoirs (<300 m) for CO_2_ capture and storage in the post-combustion process. This process is called enhanced carbon capture and storage (e-CCS). In this process, it is necessary to use a nano-modifying agent to improve the chemical-physical properties of geological media, which allows the performance of CO_2_ selective adsorption to be enhanced. Therefore, this study presents the development and evaluation of carbon sphere molecular nano-sieves (CSMNS) from cane molasses for e-CSS. This is the first report in the scientific literature on CSMNS, due to their size and structure. In this study, sandstone was used as geological media, and was functionalized using a nanofluid, which was composed of CNMNS dispersed in deionized water. Finally, CO_2_ or N_2_ streams were used for evaluating the adsorption process at different conditions of pressure and temperature. As the main result, the nanomaterial allowed a natural selectivity towards CO_2_, and the sandstone enhanced the adsorption capacity by an incremental factor of 730 at reservoir conditions (50 °C and 2.5 MPa) using a nanoparticle mass fraction of 20%. These nanofluids applied to a new concept of carbon capture and storage for shallow reservoirs present a novel landscape for the control of industrial CO_2_ emissions.

## 1. Introduction

Global warming is still a controversial topic worldwide. Its origin is mainly attributed to (1) Planet Earth’s natural cycles, which cause changes in the global temperature, and (2) human activities (anthropogenic global warming), which cause the deterioration of nature and a greenhouse effect. This has allowed a better compression of both research lines and how they can be connected. Anthropogenic global warming has gained importance in recent years because of scientific research and reports showing the relationship between the increase in gas emissions and the increase in global temperature [1,2,3,4,5]. There are different greenhouse gases, but global warming is mainly related to large amounts of carbon dioxide emissions [6,7,8,9,10,11], of which about 78% is generated by industry through combustion processes [8,12,13]. Industrial emissions (flue gas) are composed of 15%–20% CO_2_, 5%–9% O_2_, traces of other gases, and the N_2_ balance [14,15]. However, current methods are not good enough to control CO_2_ emissions [16,17]. The most frequently used method at the industrial level is absorption, but this has significant problems related to solvent regeneration, corrosion, and a high energy consumption, among others [18,19,20,21]. Considering this, many institutions around the world are promoting the exploration of other options for stabilizing and controlling CO_2_ emissions in future years [6,7,8,22]. The International Energy Agency presents different options that could stabilize the emissions by 2035, such as enhancing the energy efficiency (50%), alternative energies (16%), biofuels (4%), nuclear energy (9%), and the carbon capture and storage process (CCS) (22%).

Therefore, CSS has been promoted and evaluated around the world in the last decades [13,21,23,24]. This method consists of three main stages: (1) CO_2_ separation from flue gas, (2) CO_2_ transport to the geological store, and (3) CO_2_ injection into deep geological storage sites (more than 300 m [21]). In this case, supercritical CO_2_ is stored through the free space in the geological medium. Nevertheless, industrial massification of the CCS process still has different technical and economic challenges [13,21], mainly in the first and second stages (separation), which can consume about 70%–80% of the total CCS project costs [25,26,27].

Accordingly, Rodriguez et al. [24] presented an enhanced CCS process (e-CCS) based on nanotechnology to avoid separation, in which transport is avoided as CO_2_ is separated and stored underground in-situ in shallow geological reservoirs (<300 m). There are many geological media that could be used for the e-CCS process. Natural geological media, such as gas/oil reservoirs, could be perfect for CO_2_ storage due to their physical conditions, such as nature seals, which could prevent possible operational problems in the future. In sandstone reservoirs, the adsorption capacity is less than 0.0013 mmol g^−1^, which can be improved in terms of the selectivity and uptake by surface modifications of the rock [24]. In this case, CO_2_ remains at gas conditions and, through the injection of carbon nanospheres, selective adsorption controls the capture and storage processes [24]. On the other hand, the spherical structure and nanometric size allow the conservation of the porous structure of the geological media, avoiding the clogging of pores, which prevents operational problems due to formation damage and the associated loss of injectivity [28,29]. One great advantage of these nanospheres is that these materials are synthesized with specific chemical-physical characteristics for CO_2_ adsorption process [30], but they are not selective for carbon dioxide, as they show a high adsorption capacity towards other molecules in the flue gas stream, such as nitrogen.

It is worth mentioning that carbon nanomaterials have many advantages related to their high surface area, tunable porous texture, and easy surface functionalization [20,31,32,33]. Additionally, these materials can be obtained from carbon biomass precursors, resulting in a low cost [34,35].

Although there are reports of other materials with high adsorption capacities (for example, Mg-MOF74, with a CO_2_ adsorption capacity of up to 10 mmol g^−1^ at 25 °C and 0.01 MPa [36]), their stability and physical properties may limit their application at the reservoir level. In the same way, 2D and 3D nanometric structures (nanofibers, nanosheets, and nanotubes) have been reported for CO_2_ adsorption, where the capacity at a high pressure can be greater than 4–5 mmol g^−1^. Nevertheless, these materials have limitations for geological application [20,32,37,38,39]. Hence, due to their nanometric size and shape, nanospheres are the best option for e-CCS. However, the literature is very limited for these spherical nanomaterials applied to the CO_2_ capture process evaluated at atmospheric pressure (average CO_2_ adsorption capacity is between 2 and 4 mmol g^−1^) and high pressure (average CO_2_ adsorption capacity is greater than 3–5 mmol g^−1^) [20,24,37,40,41,42].

In this sense, agriculture is an integral part of the economy of many countries. In Colombia, agricultural activities are developed in many rural areas that have been vulnerable to violent conflict [43,44]. These areas are significant producers of cane, rice, corn, and fruits, among other products, but several production chains have problems related to waste management. Organic waste or biomass, in some cases, represents social, health, and environmental problems [43,44,45]. Nevertheless, these organic residues have the potential to be used as carbon precursors for micro/nanomaterial production, especially those that have a high carbohydrate content [20,30,35]. Therefore, the International Sugar Organization estimates that about 110 countries produce sugar, and production from cane represents nearly 80% of global sugar production. In general, 100 tons of sugar cane will yield 10–11 tons of sugar and 3–4 tons of molasses [46], which makes it an excellent candidate for the synthesis of carbon-based nanomaterials.

To the best or our knowledge, in the scientific literature, there is no research related to the use of carbon nanospheres synthesized from biomass residue with molecular sieve-type behavior for improving the natural selectivity for CO_2_ uptake.

Therefore, the main objective of this study was to synthesize and characterize carbon sphere molecular nano-sieves (CSMNS) from cane molasses. Nanofluids were prepared using the obtained CSMNS at different loads for impregnating the rock surface at 10% and 20%, in order to obtain a considerable change in the adsorption capacity of sandstone. Then, the materials were evaluated in the CO_2_/N_2_ adsorption processes at different pressures to mimic the e-CCS process conditions. As the main novelty, carbon nanospheres obtained from cane molasses behave like molecular sieves, which allows a natural selectivity towards CO_2_ in mixtures of CO_2_/N_2_ under different pressure and temperature conditions of an e-CCS process. The material could be considered a molecular sieve, due the fact that it only allows the entry of CO_2_ molecules within its porous structure, which is evidence of a high adsorption capacity, while it does not allow the entry of N_2_ molecules, thus exhibiting a poor adsorption capacity.

## 2. Materials and Methods

The carbon sphere molecular nano-sieves (CSMNS) were synthesized by means of a hydrothermal method using latex nanospheres as a template to obtain “hollow” spheres. The synthesized nanostructures were carbon micro/nanostructures obtained from cane molasses as a carbon precursor (CN.RON), and carbon nanostructures obtained from resorcinol/formaldehyde as a carbon precursor (CN.POL), which was synthesized to conduct a comparison with the adsorption performance of CN.RON. The synthesis from cane molasses was carried out by employing the H_2_O/molasses mass ratios of 1800:1 (CN.RON1) and 3600:1 (CN.RON2). The materials were characterized to select the best material, considering important parameters in relation to the e-CCS process, such as the smallest size, adsorption capacity, and lower technical and economic cost [24]. The best material was impregnated over Ottawa sandstone, which was used as a porous medium. The CO_2_ adsorption was evaluated at atmospheric pressure and 0 °C, as well as at high pressure and temperatures of 25 and 50 °C (reservoir conditions).

### 2.1. Materials and Reagents

Cane molasses mainly consist of water, carbohydrates, protein, and fibers. The composition of the molasses employed for the nanomaterial synthesis is shown in Table 1.

Furthermore, for synthesis, cleaning, drying, and carbonization, the following chemicals were used, all from Sigma–Aldrich (USA): Styrene (97%), acetoacetoxyethyl methacrylate (97%), sodium peroxydisulfate (97%), formaldehyde (37%), resorcinol (≥99%), Pluronic F127, acetone (99.9%), and ethanol (99.5%). N_2_ (high purity, grade 5.0) was used for carbonization.

### 2.2. Synthesis of Nanomaterials

The materials developed were latex spheres as templates for carbon spheres from resorcinol/formaldehyde (CN.POL) and carbon spheres from cane molasses (CN.RON).

#### 2.2.1. Synthesis of Latex Sphere Templates

Latex was selected as the template considering that it is composed of carbon-based molecules, which are not completely removed from the nucleus after carbonization at 500 °C. This results in a different porous internal structure, which can be modified with other compounds to enhance CO_2_ capture and storage.

The process for obtaining the latex template was based on the method proposed by Agrawal et al. [47]. For this, water (85 g), styrene (9.5 g), and acetoacetoxyethyl methacrylate (0.5 g) were mixed and charged into a double-wall glass reactor, using a mechanical stirrer, temperature controller, and nitrogen inlet. First, N_2_ was bubbled through the reaction media for 30 min. After that, the reaction temperature was increased to 70 °C, and aqueous sodium peroxydisulfate (SP) solution (mass relation SP/water of 0.03:1) was added to reaction media to start the polymerization. The reaction was carried out at 70 °C for 24 h. The polystyrene spheres were dried under dynamic vacuum.

#### 2.2.2. Synthesis of CN.RON

The process was adapted from White et al. [35], using the latex template and defining particular concentrations of reagents and carbon precursor. The initial water:carbon precursor mass ratio was 1800:1 and the latex/carbon precursor mass ratio was 1:10. The latex nanoparticles were placed in water with 0.05 mL of Span 80 to disperse hydrophobic particles in the aqueous medium, at 25 °C for 4 h and 200 rpm. After that, the carbon precursor was added to the system, which was followed by stirring for 30 min. This solution was put in a hydrothermal reactor (Techinstro, Nagpur, India) with a capacity of 200 mL, at 180 °C for 24 h. After the reaction, the carbon gels were filtered and washed with excess deionized H_2_O.

Later, gels are washed with acetone for 3 days, to permit the exchange of water molecules and preserve the porous structure during carbonization. Finally, the obtained polymer was dried at 120 °C for 12 h and carbonized under N_2_ flow at 60 mL min^−1^ and 500 °C (1 °C min^−1^) for 6 h, using a tubular furnace (Thermo Fisher Scientific, Waltham, USA). CN.RON1 was obtained. After that, the same procedure was carried out at a lower molasses concentration. The cane molasses was diluted at a mass ratio (H_2_O/carbon precursor) of 3600:1 to obtain smaller particle sizes. In this case, CN.RON2 was obtained.

#### 2.2.3. Synthesis of CN.POL

The general procedure was adapted from Fang et al. [48]; the carbon precursor and some operating conditions were changed to obtain a nanometric size and high surface area. The original method uses phenolic resol as the carbon precursor without a template, but the material yield is less than 10%. For this reason, resorcinol was used as a carbon precursor. Pluronic F127 was employed as a soft template, in order to obtain a porous structure.

Initially, a solution of resorcinol/formaldehyde (R/F) (molar ratio of 1:2), Pluronic F127 (concentration of less than 10^−7^ mol L^−1^), and deionized water (molar relation water/resorcinol ratio of 5556:1) was stirred at 200 rpm, 25 °C, for 18 h. Parallel to this, the latex nanoparticles were placed in water with 0.05 mL of Span 80 to disperse hydrophobic particles in the aqueous medium, at 25 °C for 4 h and 200 rpm. After that, the carbon precursor was added to the system, which was followed by stirring for 30 min. The solution was placed in a hydrothermal reactor (Techinstro, Nagpur, India) with a capacity of 200 mL, for 24 h at 130 °C. The obtained polymer was cleaned with acetone for 3 days to remove the water inside the porous structure. After that, the material was carbonized in N_2_ at 700 °C (1.5 °C min^−1^) for 4 h, using a tubular furnace (Thermo Fisher Scientific, Waltham, USA).

### 2.3. Impregnation of Sandstones

The sandstone was modified to enhance the rock surface with the nanoparticles and increase the chemical-physical condition and molecular interactions [24,49]. According to Rodriguez et al. [24], the minimum nanoparticle percentages needed to enhance the CO_2_ adsorption capacity are 10% and 20%. Considering this, in this work, Ottawa sandstone (SS) was impregnated with CN.RON2 at mass fractions of 10% and 20% by the immersion and soaking method [49]. The nanofluid was developed by dispersing the nanoparticles in deionized water, which was sonicated at 40 °C for 4 h. The SS was introduced into the nanofluid at 60 °C for 24 h. This method mimicked the reservoir conditions. The modifying SS was dried at 110 °C for 12 h.

### 2.4. Characterization of Materials

Chemical-physical properties can be evaluated by different techniques. For e-CCS, it is essential to obtain nanomaterials with the smallest size, a spherical structure, and a high surface area. The details of characterization are presented below.

#### 2.4.1. Size and Structure

Scanning electron microscopy (SEM) was used to obtain the dry particle size, size distribution, and morphology of CN.RON and SS. Transmission electron microscopy (TEM) was used to analyze the “hollow” structure. The analysis was carried out using a JSM−7100 emission electron microscope (JEOL, Nieuw-Vennep, The Netherlands), a GEMINI-LEO1530 VP FE-SEM emission electron microscope (Carl Zeiss, Cambridge, UK), and a Tecnai F20 Super Twin TMP transmission electron microscope (FEI, Hillsboro, USA).

A NanoPlus-3 zeta/nanoparticle analyzer (Micromeritics, Norcross, USA) at 25 °C was used to obtain the mean particle size of nanoparticles dispersed in fluid (dynamic light scattering (DLS)), where the particles were hydrated and interacted with each other [24]. In this case, the nanomaterials were dispersed in water or ethanol (10 mg L^−1^) and sonicated for 6 h before analysis.

#### 2.4.2. Porous Structure

Materials were characterized by N_2_ and CO_2_ adsorption at −196 and 0 °C, respectively, using Autosorb adsorption equipment (Quantachrome Instruments, Anton Paar Quanta Tech, Boynton Beach, USA). The total adsorbed volume (*V*_0.95_) is the physisorbed N_2_ volume at a relative pressure of *P*/*P*_0_ = 0.95. The surface area (*A*_BET_) was obtained by the Brunauer–Emmett–Teller (BET) model [50]. The micropore volume (*V*_mic_) and average pore size (*L*_0_) were obtained by the Dubinin–Radushkevich model [51,52]. The mesopore volume (*V*_meso_) was obtained by the Barrett–Joyner–Halenda (BJH) model [53,54].

#### 2.4.3. Chemical Composition

The chemical characterization was carried out by carbon, hydrogen, oxygen, and nitrogen (CHON) analysis for nano/micromaterials using a CE 440 elemental analyzer (Exeter Analytical Inc, Chelmsford, USA). Moreover, Fourier transform infrared spectroscopy (FTIR) for carbon materials and sandstone was performed using potassium bromide at a KBr/material ratio of 30:1 (% w/w) and at 25 °C. For this, an IRAffinity-1S FTIR spectrometer was used (Shimadzu Scientific Instruments, Columbia, USA). Thermogravimetric analysis (HP TGA 750, TA Instruments, New Castle, USA) was used to analyze the impregnation percentages of nanoparticles on sandstone. For this, the samples were burned at 800 °C (10 °C min^−1^) under an air atmosphere.

#### 2.4.4. Dispersion of Nanoparticles in Aqueous Media

A NanoPlus-3 zeta/nanoparticle analyzer (Micromeritics, Norcross, GA, USA) was used to analyze the surface charge of particles and their dispersion stability at 25 °C (electrophoretic light scattering (ELS)). Nanoparticle/water suspensions were prepared with a pH adjusted to between 2 to 12 by adding solutions of 0.1 mol L^−1^ HCl or 0.01 mol L^−1^ NaOH, and then subjected to analysis. The zeta potential was calculated using the Smoluchowski equation, derived from the calculation of the Doppler effect [55,56,57].

### 2.5. Adsorption Tests

The HP TGA 750 thermogravimetric analyzer (TA Instruments, New Castle, USA) was used to evaluate the performance adsorption of CN.RON2, sandstone, and impregnated sandstone (with a mass fraction of 10% and 20% of CN.RON2) at 25 and 50 °C and high pressure from 0.03 to 2.5 MPa for CO_2_ and N_2_. Initially, the material was cleaned by vacuum to remove gases and humidity. For this, the thermogravimetric analyzer had a vacuum pump coupled with an oven, in which the sample was placed. The moisture and adsorbed gases were removed from the surface and immediately afterwards, the adsorption process was carried out. The system was operated at 0.0025 MPa and 120 °C for 12 h. This equipment used magnetic levitation top-loading balance, which generated a uniform electromagnetic field. The sample weight was proportional to the current required to maintain the balance position. Additionally, this system allowed a highly sensitive analysis. The contribution of the buoyancy effect was manually subtracted from the data using blank tests carried out in the same conditions, but with an empty sample holder [24]. The amount of each material was around 15 mg for nanoparticles, and 40 mg for sandstone and impregnated sandstone, in order to produce enough total surface area for adsorption.

The isotherms were fitted with the Sips and Toth models, which take into account multilayer adsorption. The models are presented in Table 2 [24,34,58,59]. *K*_S_ and *K*_T_ represent the adsorption equilibrium constants for the Sips and Toth models, respectively, and the *n* and *t* parameters indicate the heterogeneity of the system for the Sips and Toth models, respectively. *N*_ads_ (mmol g^−1^) is the adsorbed amount, *N*_m_ (mmol g^−1^) is the adsorption capacity at equilibrium, and *P* (kPa) is the equilibrium pressure. The heterogeneity may originate from the solid structure, the solid energy properties, or the adsorbate [34].

## 3. Results and Discussion

### 3.1. Characterization of Nanomaterials

This section is divided into three main parts: the size and morphology, chemical composition, and porous structure.

#### 3.1.1. Size and Morphology

The morphology and size distribution of the carbon materials obtained from molasses at a mass ratio of H_2_O/carbon precursor of 1800:1 (CN.RON1) and 3600:1 is presented in Figure 1a,b, respectively. CN.RON1 is composed of spheres (size > 3 μm), but other carbon structures are presented and may be related to the other original components of molasses or latex template without carbon cover. Due to the growth of the CN.RON1 particles, a hollow structure cannot observed. In addition, the concentration of carbon precursors can be high, which allows a high growth of spheres. This behavior is similar to that reported by Li et al. [49,52], Bai et al. [50], and others [30,60,61,62,63]. CN.RON2 presents a nanometric size and aspherical structure (Figure 1b). Additionally, CN.RON2 presents a smaller size than the latex template, which could be due to the carbohydrate aggregation speed being greater than the coating speed over the template. Moreover, the amount of template is lower for CN.RON2 synthesis (mass ratio of template:carbon precursor of 1:10). It is important to mention that the hydrophobic template is dispersed in the aqueous medium by means of a surfactant.

The latex template is presented in Figure 1c, in which nanospheres of a homogeneously distributed size can be observed, but with a bit of aggregation between particles. For this reason and taking into account its hydrophobic character, the use of surfactants is necessary.

To have a point of comparison, the material obtained from a biomass resource (CN.RON2) was compared to particles synthesized from resorcinol/formaldehyde (CN.POL). The obtained spherical particles present a relatively homogeneous size distribution (Figure 1d). Furthermore, CN.POL does not present aggregation, such as latex nanoparticles, which indicates the excellent dispersion of latex nanoparticles before the reaction. As expected, the particles do not have a completely hollow core due to the carbon content of the latex compound (Figure 1e), which is in agreement with the study by White et al. [35].

To reduce the particle size, a second synthesis process was performed with a mass ratio of H_2_O/carbon precursor of 3600:1. The size was analyzed using dynamic light scattering (DLS). Table 3 presents the mean particle sizes of CN.RON and CN.POL. The diameter allows the interactions of particles or aggregates to be analyzed.

The hydrodynamic diameter was obtained for particles in water (at pH 5.8) and ethanol (at pH 7). The interfacial tension was lower in the alcohol/particles system, and the aggregate size was smaller in ethanol (Table 3). The aggregation was higher at pH values between 4.7 and 6.8 (Figure 2), for which the zeta potential was close to zero. If the zeta potential is high (negative or positive) at the pH evaluated, there is high electrostatic repulsion, allowing stable particles. A nanofluid based on water could be the most economical way to introduce nanomaterial into real porous media. In this system, the particles could be aggregated (Figure 2). Therefore, an analysis of the pore size distribution of the geological formation must be contrasted with the particle size of nanofluid for preventing formation damage.

When particle aggregates are big enough, they can precipitate. This effect happened for CN.RON1, which has a considerable particle size (Figure 1) and in an aqueous medium (at pH between 5 and 7), tends to increase. Hence, the hydrodynamic diameter of CN.RON1 in water is greater than the limit of detection of the equipment (10 μm). For CN.RON1 in ethanol, the measurement was unstable, possibly due to the mixing of dispersed particles and aggregates that exceeded the size limit for detection. The hydrodynamic diameter was smaller for CN.RON2, as was expected, and the particle size could be less than 100 nm, but in the aqueous system, the particles were hydrated and interacted with other particles, thus increasing a bit in size. This agrees with the result obtained by means of scanning electron microscopy (SEM) (Figure 1b).

According to the size of the CN.RON1 material, this was discarded for further evaluation.

#### 3.1.2. Chemical Composition

Figure 3 shows the FTIR spectra for the two carbon sphere samples synthetized (CN.RON2 and CN.POL). Around 3600 cm^−1^, the contribution of the O-H from ambient water can be seen. This band shows the stretching vibrations (3600–3100 cm^−1^) due to the hydroxyl groups and chemisorbed water on the surface. The band at 3600–3100 cm^−1^ indicates the asymmetry and hence the presence of hydrogen bonds. The bands at 2844 and 2925 cm^−1^ are commonly used for describing C–H stretching. However, the CN.RON2 spectrum shows that these bands may overlap with the OH stretching band (3600–3100 cm^−1^) and the aromatic ring bands and double bond (C = C) vibrations (1650–1500 cm^−1^) [64,65]. In addition, CN.RON2 had a considerable nitrogen content (Table 4), which suggested that an activated carbon surface forms surface complexes with the presence of nitrogen surface functional groups, which are desirable for CO_2_ adsorption [65].

The elemental analysis (CHON) was carried out using the CE-440 analyzer for both CN.RON2 and CN.POL (Table 4). The oxygen value was measured independently. CN.RON2 presents a nitrogen (N) content that is considerably higher than that of the CN.POL sample. It should be taken into account that the material comes from biomass and has not undergone any additional treatment [46].

#### 3.1.3. Porous Structure

The adsorption isotherms of N_2_ and CO_2_ are presented in Figure 4, and the parameters obtained from these isotherms are shown in Table 5. For CN.POL, it is possible to obtain an N_2_ adsorption isotherm. Nevertheless for CN.RON2, the adsorption capacity is practically negligible; regardless of the operating conditions used, its A_BET_ is 2 m^2^ g^−1^ and pore volume is practically 0. On the other hand, the CO_2_ adsorption capacity is 3 mmol g^−1^ for CN.RON2, which is very similar to CN.POL, and has an A_BET_ of 585 m^2^ g^−1^ and a well-developed micropore structure. From the CO_2_ isotherm, it can be ascertained that CN.RON2 has a micropore volume of 0.22 cm^3^ g^−1^. Therefore, the CO_2_ molecule size means that N_2_ cannot enter (molecular size of 4 Å for CO_2_ and 3.68 Å for N_2_). Additionally, the strong molecular interactions directly impact the interaction adsorbate/adsorbent, due to the polarizability and quadrupole moment, which benefits molecular interactions of CO_2_/CN.RON2 (polarizability is 2.93 × 10^−40^ C^2^ m^2^ J^−1^ for CO_2_ and 1.97 × 10^−40^ C^2^ m^2^ J^−1^ for N_2_; and quadrupole moment is 13.4 × 10^−40^ C m^2^ for CO_2_ and 4.7 × 10^−40^ C m^2^ for N_2_) [66]. If the molecular polarity is higher, then the adsorption velocity is faster through entry into the pore [66]. Although the differences found are considerable, some other effects could be responsible for this, such as textural effects, which combined with the pore size could, give the molecular sieve behavior. For determining this, it is necessary to evaluate and corroborate the behavior based on the adsorption isotherms at other operating conditions (pressure and temperature), which is in agreement with the reports by Mohamed et al. [67], Buonomenna et al. [68], and Foley et al. [69].

On the other hand, CN.RON2 has a considerable CO_2_ adsorption capacity (similar value for both materials, CN.RON2 and CN.POL) and competitive values compared to the values reported in the literature for carbon materials at the same conditions [20,37,58]. The microporous volume of CN.RON2 (from the CO_2_ adsorption at 0 °C) is not larger than that of other material (CN.POL), although its high adsorbed amount at the conditions evaluated showed a good performance, which could be associated with a higher affinity, despite its microporous volume.

Based on the data of N_2_ and CO_2_ adsorptions obtained at −196 and 0 °C, respectively, CN.RON2 could have molecular sieve characteristics due to its null nitrogen adsorption. Therefore, in the next stage, adsorption tests were performed at different operating conditions to rule out textural effects.

### 3.2. High-Pressure Adsorption Analysis

#### 3.2.1. Adsorption on CN.RON2 and CN.POL Nanomaterials

Figure 5 presents the adsorption isotherms for CN.RON2 at temperatures of 25 and 50 °C for pressures up to 2.5 MPa for CO_2_ and N_2_, and the fit with Sips and Toth models. Concerning the adsorption of CO_2_ at 25 and 50 °C, as expected, the adsorbed amount is higher for the lower temperature, with a difference of 1.32 mmol g^−1^ (at 2.5 MPa), due to the exothermic character of adsorption, which reduces the interaction forces between the adsorbate and adsorbent. At higher temperatures, there is more internal energy, and hence more CO_2_ molecules in the gas phase, reducing the uptake adsorbed. The effect of temperature is in agreement with that reported in the literature [70]. The adsorption isotherms of CO_2_ obtained for the CN.RON2 sample show a type I behavior considering IUPAC classification [71,72,73], where is possible to observe that the adsorbed amount of CO_2_ increases as the absolute pressure increases. The greatest increase of the CO_2_ adsorption capacity (approx. 70%) occurs at low pressure values (*p* < 0.5 MPa), and later, the pressure increase (*p* up to 2.5 MPa) allows the amount adsorbed at equilibrium (30% remaining) to be reached, which is presented in Figure 5. This phenomenon is common for microporous materials where the adsorption is due to the filling of pores for a monolayer adsorption mechanism [71,72].

At pressures higher than 0.2 MPa, the increase of the amount adsorbed is gradual. For pressures <0.2 MPa, the Henry region is found, showing a higher affinity for the CO_2_ as the temperature decreases. Meanwhile, the nitrogen adsorption is null regarding the CO_2_ adsorption for CN.RON2. This behavior is typical for sieve samples where the materials do not have an affinity for the nitrogen.

On the other hand, CN.RON2 could be considered as a carbon spherical molecular nano-sieve, which has not been reported in the literature. The nano-sieve behavior of CN.RON2 can be attributed to its absorbing behavior, allowing only CO_2_ molecules to selectively enter its porous structure; for this reason, a high CO2 adsorption capacity exists. On the other hand, the N_2_ adsorption capacity is negligible, even at high-pressure conditions (<0.05 mmol g^−1^), due to the characteristic of the CN.RON2 sample, which is a nano-sieve and non-selective for nitrogen. This behavior agrees with that reported by Mohamed et al. [67], Buonomenna et al. [68], and Foley et al. [69].

CN.POL has a CO_2_ adsorption capacity of 4.1 mmol g^−1^ at 3 MPa and 50 °C, and its nitrogen adsorption at the same conditions is much higher in comparison to that obtained for CN.RON2 (2.6 mmol g^−1^). Hence, the CN.POL was discarded for sandstone modification as there was no evidence of CO_2_ selectivity.

In this study, the Sips and Toth models were used for describing the behavior of the adsorption isotherms at different temperatures. The Sips model provides the best fit with the CN.RON2 for the two temperatures evaluated. It is worth mentioning that this model describes the adsorption phenomena based on the monolayer adsorption, i.e., a molecule can only occupy one adsorption site during the adsorption process [34]. The CO_2_ adsorption parameters can be found in Table 6, and achieved good fits with R^2^ > 0.99 (Sips model) and R^2^ > 0.98 (Toth model). The N_2_ isotherms present a poor affinity and adsorbed amount, and the Sips and Toth models do not have a good fit, compared to the fit of CO_2_ isotherms (R2 ≤ 0.95).

#### 3.2.2. CO_2_ Adsorption on Sandstone before and after Modification with CN.RON2 Nanoparticles

The impregnation process on the rock of the reservoir allowed a sandstone surface with a homogeneous nanoparticle distribution to be obtained, as can be seen in Figure 6. Figure 6a,b present the SEM images of the sandstone before and after impregnation with the CN.RON2 (mass fraction of 20% of CN.RON2). The size of the aggregate is less than 200 nm, which indicates a good dispersion of the nanoparticles after the impregnation process. It is worth mentioning that after one year of preparation, the impregnated sandstone does not present disaggregation from the surface, showing the great stability of the CN.RON2 on the rock surface of the reservoir.

The TGA analysis for SS-10 and SS-20 presents variation of the impregnation percentage. The real impregnation percentages are 8.6% and 20.8%, respectively (see Appendix A).

According to Rodriguez et al. [24] and FTIR analysis, the sandstone composition is mainly silica, which interacts with other molecules like an acid. For this reason, the interaction with CO_2_ is affected, considering the fact that the CO_2_ molecule also interacts like an acid with other molecules [74,75]. Therefore, the interaction between the rock surface and CO_2_ is weak [24]. On the other hand, sandstone has a low specific area, being less than 1 m^2^ g^−1^. Both its chemical composition and surface area are responsible for the low CO_2_ adsorption capacity. The sandstone was evaluated in previous work [24] (A_BET_ is 0.4 m^2^ g^−1^) and its CO_2_ adsorption capacity was less than 0.0013 mmol g^−1^ at 0 °C and atmospheric pressure. The CO_2_ adsorption over the sandstone impregnated with CN.RON2 nanoparticles at mass fractions of 10% and 20% (CN.RON2), at 25 and 50 °C, is shown in Figure 7. The CO_2_ adsorption capacity is considerably higher than the adsorption capacity of the sandstone without surface modification.

Table 7 presents the parameters related to Sips and Toth models, which achieved good fits (R^2^ is between 0.96% and 0.99%). The system affinity (adsorbate/adsorbent) is represented by adsorption equilibrium constants (K_S_ and K_T_). The system heterogeneity is represented by n and t parameters, which could be generated by the energy properties of the solid structure [59].

At reservoir conditions (50 °C and 2.5 MPa), the increment percentage could be 73% at 20% of CN.RON2 (Figure 8).

To account for the synergistic effect of the proposed modification, the theoretical amount adsorbed was calculated as the sum of the individual effect of the sandstone and the nanoparticles. A comparison of the theoretical and experimental adsorption capacities is shown in Table 8. It can be observed that the experimental adsorption capacity is higher than the theoretical adsorption capacity, which indicates a synergistic effect. The nanoparticle comes from a biomass by-product precursor in which other components may be interacting with CO_2_ molecules and/or geological media. Moreover, this effect could be related to the CO_2_ quadrupole moment, which allows CO_2_–CO_2_ interaction at high-pressure conditions and surface interactions.

The carbonization of polymer precursors usually produces carbon-based molecular sieve membranes at high temperatures, involving complex reaction processes [76,77]. Although research on carbon molecular sieve membranes has been conducted since the 1990s [76,77,78], there is uncertainty about the formation of membranes in this type of systems [76,77]. Hence, this work could represent a research line for various applications in which the spherical shape and nanometric size are an advantage, as in the e-CCS [79,80]. Therefore, the latex template could interact with specific cane molasses components, such as fibers, carbohydrates, and proteins. The reported structures are diverse, but to the best of our knowledge, there are no reports on nanospheres that have this type of molecular sieve behavior. Moreover, it should be remarked that the CO_2_ adsorption capacities reported in the present work are competitive, and in some cases, superior to those found in the literature, including materials with a more complex synthesis process or more stages [66,69,78,80,81]. As mentioned above, the results of the CO_2_ adsorption capacity for nanospheres is limited, and the average CO_2_ adsorption capacity could be between 3 and 5 mmol g^−1^ [20,24,37,40,41,42].

The e-CCS process was first proposed in our previous work [24]. For this reason, it is not possible to compare the data collected, due to the fact that no results have been presented by other authors. If the only two reported works are compared, for CN.LYS2 [24] and CN.RON2 materials, the latter material presents the best results for the e-CCS at geological conditions. For SS-20, the difference between the incremental factors of CN.LYS2 and CN.RON2 is 8% at 50 °C and 23% at 25 °C. This could be related to its characteristics as a molecular sieve.

## 4. Conclusions

This is the first report on carbon sphere molecular nano-sieves in the scientific literature. The material was synthesized from cane molasses, which is a biomass by-product. CN.RON2 enhanced the factor of adsorption capacity by 730 with a mass fraction of only 20% (CN.RON2) under realistic reservoir conditions (50 °C and 3 MPa). This increase was possible thanks to the favorable chemical composition and physical structure, which promoted natural selective CO_2_ capture and storage.

Although the synthesis process was carried out at a considerable temperature (180 °C) and hydrothermal process, compared with other synthesized materials for e-CCS (such as our research published in 2019, which analyzed N-rich carbon nanospheres synthetized by the sol-gel method at 60 °C), the high natural selectivity, adsorption capacity, and cheap carbon precursor (biomass residue) of carbon sphere molecular nano-sieves make them a highly promising material for the e-CCS process. Further characterization of the structure is needed to obtain more clarity on the molecular sieve effect. Finally, it will be important to take the molasses composition of other origins into account to ensure the reproducibility of carbon sphere molecular nano-sieves.

## Figures and Tables

**Figure 1 nanomaterials-10-00980-f001:**
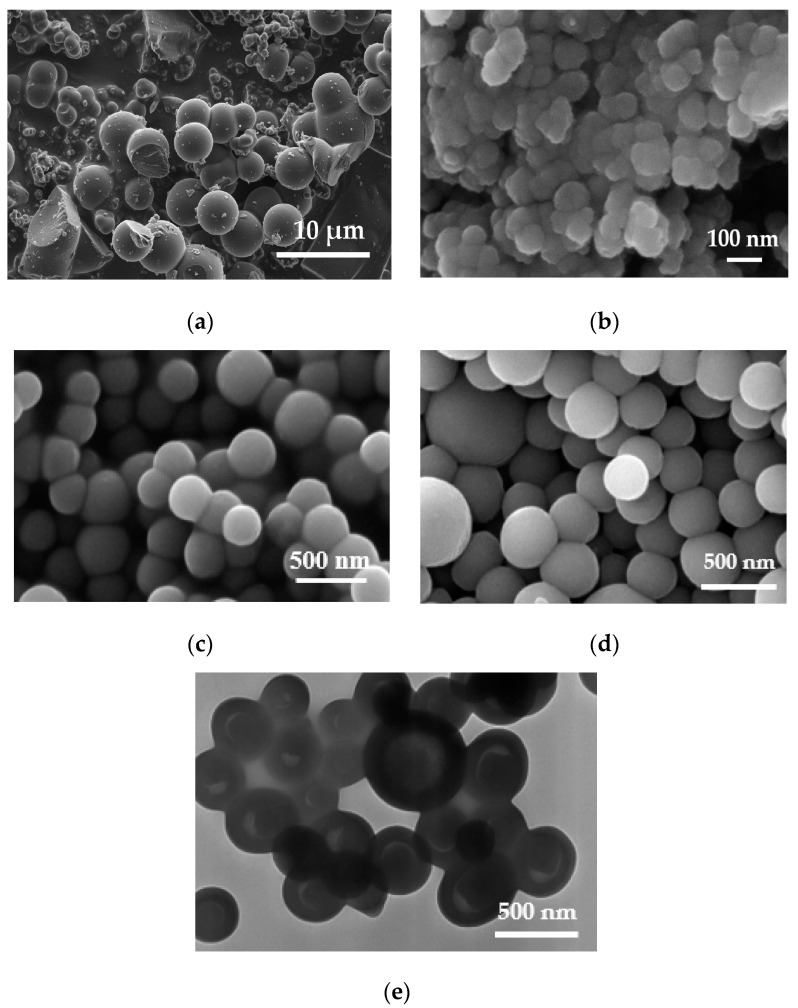
SEM images at 5 kV of carbon spheres synthesized from cane molasses (CM) at different water:CM ratios of (**a**) 1800:1 (CNRON1) and (**b**) 3600:1 (CNRON2). SEM image of latex spheres (template) (**c**). SEM (**d**) and TEM (**e**) images of carbon hollow nanospheres-CN.POL (water/resorcinol ratio of 5556:1; synthesized with a latex template).

**Figure 2 nanomaterials-10-00980-f002:**
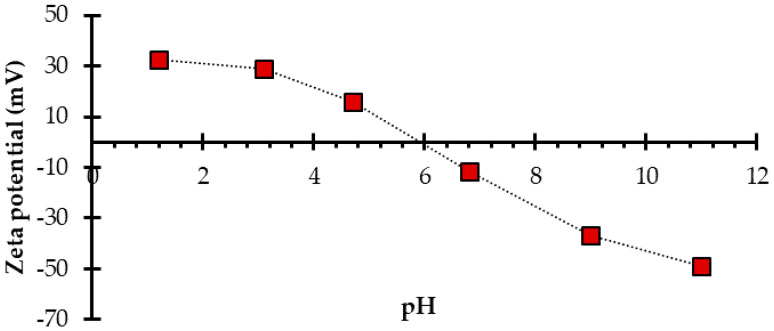
Zeta potential for carbon nanoparticles dispersed in deionized water. CN.RON2 at a mass ratio of H_2_O/carbon precursor of 3600:1.

**Figure 3 nanomaterials-10-00980-f003:**
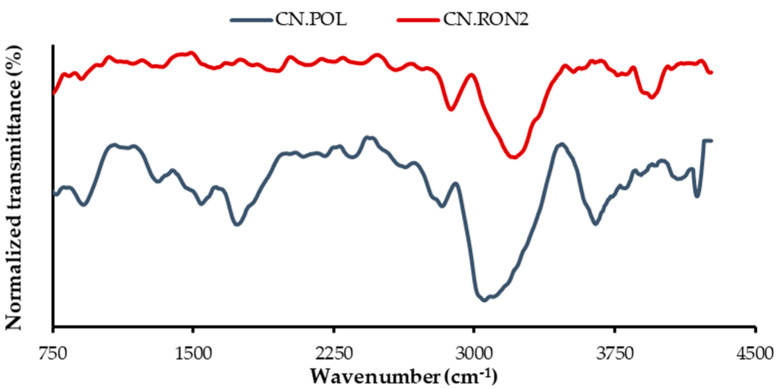
FTIR spectra of CN.POL and CN.RON2.

**Figure 4 nanomaterials-10-00980-f004:**
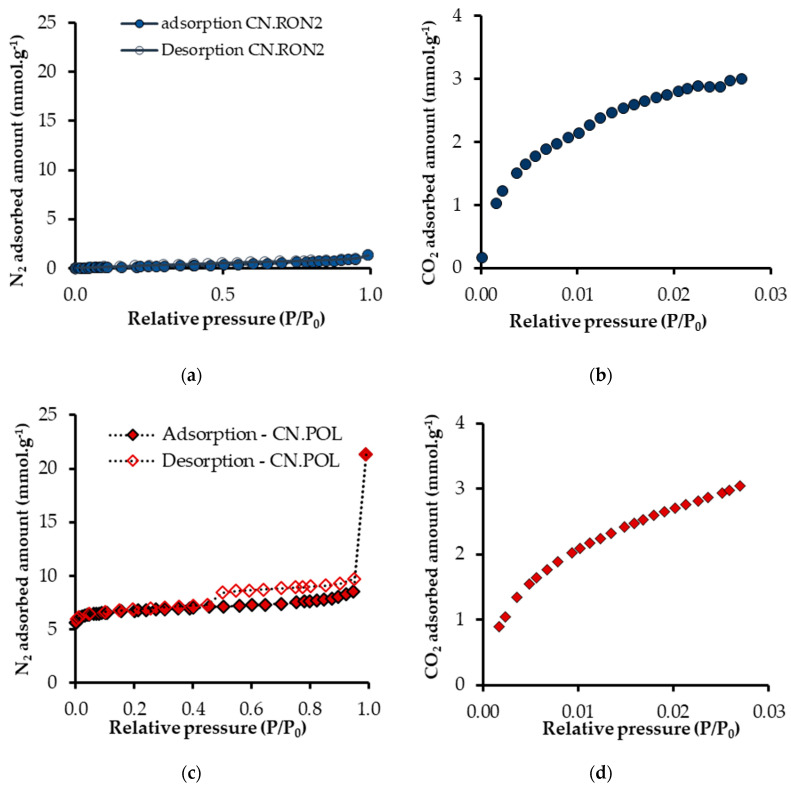
Adsorption isotherms of N_2_ (−196 °C) and CO_2_ (0 °C) for materials synthesized: (**a**,**b**) CN.RON2, and (**c**,**d**) CN.POL.

**Figure 5 nanomaterials-10-00980-f005:**
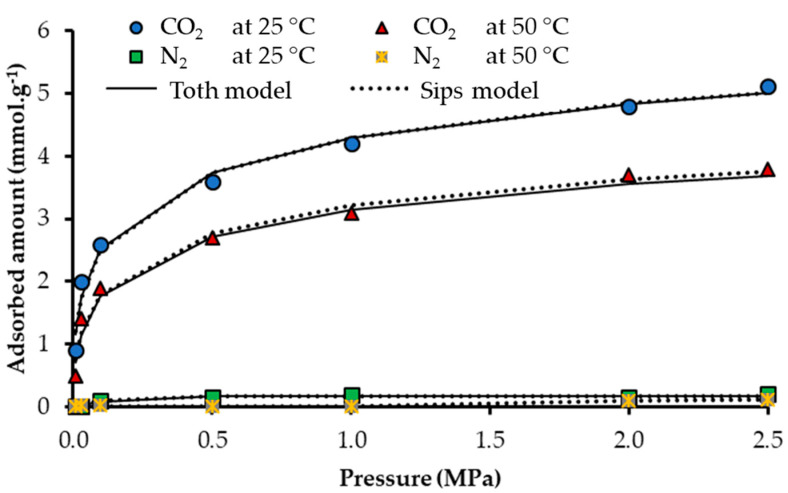
Adsorption isotherms for CN.RON2 at different conditions: 25 and 50 °C for CO_2_ and N_2_, and the fitting of Sips and Toth models. Pressure up to 2.5 MPa.

**Figure 6 nanomaterials-10-00980-f006:**
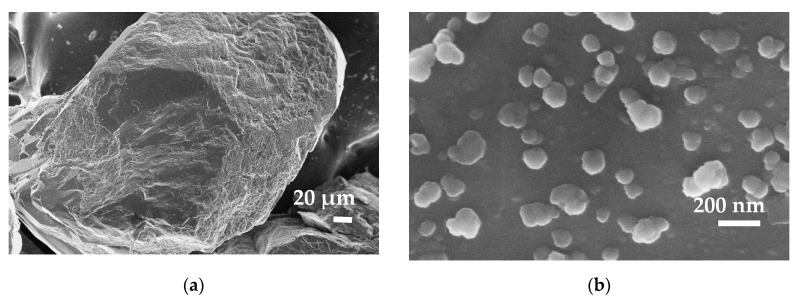
SEM images of (**a**) sandstone and (**b**) sandstone impregnated with a mass fraction of 20% (CN.RON2).

**Figure 7 nanomaterials-10-00980-f007:**
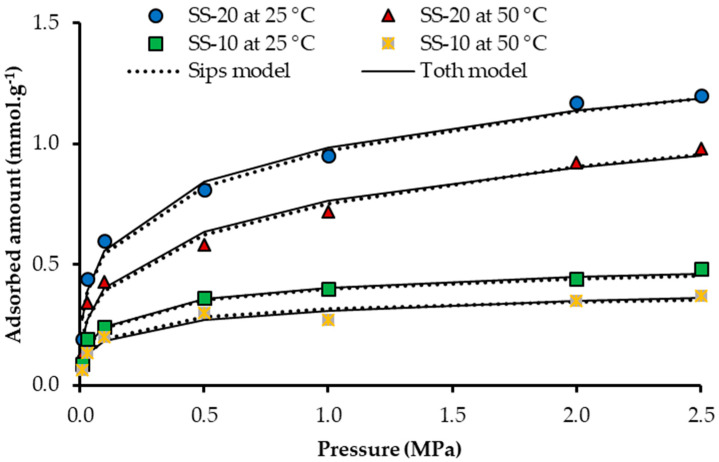
Adsorption isotherms of CO_2_ at high pressure (up to 2.5 MPa) of SS-10 (sandstone impregnated with a mass fraction of 10% CN.RON2) at 25 and 50 °C, and SS-20 (sandstone impregnated with a mass fraction of 20% CN.RON2) at 25 and 50 °C.

**Figure 8 nanomaterials-10-00980-f008:**
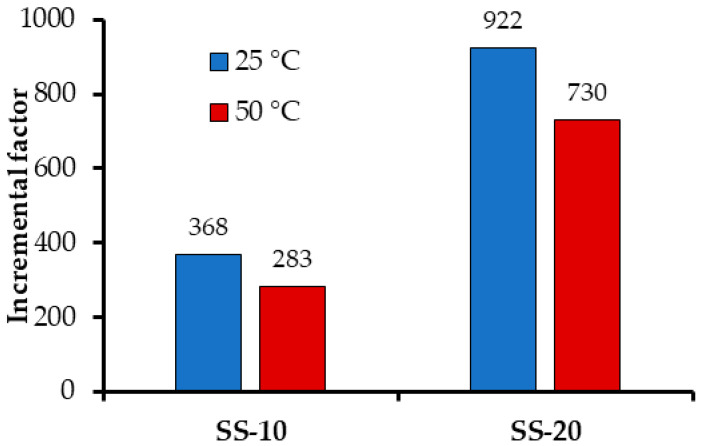
The incremental factor for CO_2_ adsorption at 2.5 MPa and 25–50 °C. Sandstone impregnated with CN.RON2 at mass fractions of 10% and 20%. Incremental factor based on the CO_2_ adsorption capacity of sandstone without impregnation.

**Table 1 nanomaterials-10-00980-t001:** Composition of cane molasses (dry base). Spanish company.

Component	Mass Percentage (%)
Carbohydrates	71.1 (sugar 56.1%)
Protein	4
Fibers	7
Na	0.5
Ca	0.6
Mg	0.3
F	0.1

**Table 2 nanomaterials-10-00980-t002:** Models for adsorption isotherms.

Model	Equations	
Sips	$N_{ads}=N_{m}\frac{{\left(K_{S}\;P\right)}^{1/n}}{1+{\left(K_{S}\;P\right)}^{1/n}\;}$	(1)
Toth	$N_{ads}=N_{m}\frac{K_{T}\;P}{{\left(1+{\left(K_{T}\;P\right)}^{t}\right)}^{1/t}}$	(2)

**Table 3 nanomaterials-10-00980-t003:** Mean particle size of nanomaterials in suspension.

Material	*d*_p_ 50 (nm) in Water (pH 5.8)	*d*_p_ 50 (nm) in Ethanol (pH 7)
CN.RON1	**-**	3650–8750
CN.RON2	256.2	143.2
CN.POL	2462	2176

**Table 4 nanomaterials-10-00980-t004:** Elemental analysis of nanoparticles synthesized with resorcinol/formaldehyde (CN.POL) and cane molasses (CN.RON2).

	% N	% C	% H	% O
CN.POL	0.03	93.2	1.9	6.7
CN.RON2	1.07	87.6	1.8	8.3

**Table 5 nanomaterials-10-00980-t005:** Parameters obtained from adsorption isotherms (N_2_ at −196 °C and CO_2_ at 0 °C) for CN.POL and CN.RON2.

	A_BET_ (m^2^ g^−1^)	V_0.95_ (cm^3^ g^−1^)	V_mic-N2_ (cm^3^ g^−1^)	V_mic-CO2_ (cm^3^ g^−1^)	V_mes_ (cm^3^ g^−1^)	L_0_ (nm)	E_ads.CO2_ (kJ mol^−1^)
CN.POL	CN.POL	0.30	0.23	0.29	0.06	0.49	33.4
CN.RON2	2	0.03	0.001	0.22	0.03	2.46	9.7

**Table 6 nanomaterials-10-00980-t006:** The Sips and Toth parameters for CO_2_/N_2_ isotherms at 25 and 50 °C. CN.RON2.

Gas	Temperature (°C)	Sips Model				Toth Model			
		Nm (mmol g^−1^)	Ks	n	R^2^	Nm (mmol g^−1^)	K_T_	t	R^2^
CO_2_	25	7.93	0.0015	2.44	0.99	10.47	0.62	0.23	0.98
	50	5.38	0.0023	2.10	0.99	6.58	0.14	0.29	0.98
N_2_	25	0.17	0.0112	0.52	0.95	0.18	0.0060	1.69	0.94
	50	0.14	0.0006	0.21	0.93	-	-	-	-

**Table 7 nanomaterials-10-00980-t007:** The Sips and Toth parameters for CO_2_ isotherms at 25 and 50 °C. Sandstone impregnated at a mass fraction of 10% (SS-10) and 20% (SS-20) of CN.RON2.

Material	Temperature (°C)	Sips Model				Toth Model			
		Nm (mmol g^−1^)	Ks	n	R^2^	Nm (mmol g^−1^)	K_T_	t	R^2^
SS-10	25	0.62	0.004000	2.11	0.99	0.73	0.22	0.30	0.99
	50	0.42	0.007000	1.76	0.96	0.74	0.68	0.23	0.96
SS-20	25	3.44	0.000050	3.11	0.96	5.38	7.45	0.14	0.97
	50	4.55	0.000006	3.08	0.98	6.61	3.42	0.13	0.97

**Table 8 nanomaterials-10-00980-t008:** Theoretical and experimental values of the CO_2_ adsorption capacity at 2.5 MPa, 25 and 50 °C, and mass fractions of 10% and 20%.

Parameter	SS-8.6		SS-20.8	
	25 °C	50 °C	25 °C	50 °C
Theoretical N_ads_ (mmol g^−1^)	0.44	0.33	1.07	0.79
Experimental N_ads_ (mmol g^−1^)	0.48	0.37	1.20	0.99
Relative difference (%)	8.0	11.2	11.1	20.0

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
