# Peer review of "Biomass-Derived Carbon Molecular Sieves Applied to an Enhanced Carbon Capture and Storage Process (e-CCS) for Flue Gas Streams in Shallow Reservoirs"

_nanomaterials, 2020, doi:10.3390/nano10050980_

Round 1

Reviewer 1 Report

The article titled: "Carbon Spherical Molecular Nano-sieves from Biomass Residues applied to Enhanced Carbon Capture and Storage process (e-CCS) for Flue Gas Streams in Shallow Reservoirs " by Elizabeth Rodriguez Acevedo et al. presents study, which aims is at the development and evaluation of carbon spherical molecular nano-sieves (CSMNS) from cane  molasses for e-CSS. This is the first report in the scientific literature of carbon spherical molecular nanosieves, due to its size and structure. The materials was synthesized by means of hydrothermal method and posteriorly characterized by several tecniques for determining the size, structure and composition...

The paper seems to be acceptable but, in my opinion, it requires some small modifications. Additionally, several questions should be answered/add or repair by the authors in detail, as many important issues are described too superficially:

  1. Line 31: “…nanoparticles mass fraction of 20%.” – isn't this percentage too much for industrial use? Do such quantities make sense and will the costs not be crazy?
  2. Keywords: - I think it's too much?
  3. Line 37: isn't it worth adding even a note about this global warming? Is it real global warming or maybe another period (in Earth's life) of rising and falling temperatures? As the authors believe - maybe it's worth mentioning?
  4. Line 50: Will this CO2 injection not create various dangers in the future? Maybe the authors could refer to it in a short sentence?
  5. Line 61: - authors should add information on how these materials do it?
  6. Line 70: - or just Colombian?
  7. Table 2 - the quality of the presentation of the designs leaves much to be desired - the authors must improve it.
  8. Fig 2 and 3 - note that the authors format the drawings differently. This must be corrected - once, for example, there is bold lines and text, other times not.
  9. Fig 3 - the spectra are of poor quality. You can mark the bands the authors write about (in fact correctly) in the drawing - because that's how the potential reader must guess where it is.
  10. You can slightly expand the summary section - so that it more reflects the work because it has little to do with it.

In conclusion, the paper seems to be acceptable but requires some small revisions. The whole layout and neatness of the paper do not leave too much objections, as it is prepared very carefully, but the quality of the discussion requires several amendments.

Please answer all my questions and comments. The work is very good and worth publishing, but some improvements should be made.

The objections presented by me do not undermine the quality of the paper, which will support in the further publishing process, certainly after careful consideration of my comments.

Author Response

Responses to the Academic Editor suggestions for the manuscript: Carbon Spherical Molecular Nano-sieves from Biomass Residues applied to Enhanced Carbon Capture and Storage process (e-CCS) for Flue Gas Streams in Shallow Reservoirs

We thank the reviewer for her/his effort made for securing a prompt review of our manuscript and the opinion of publication after correction. Also, we thank for all revisions, all of them were included in the manuscript, and the text was improved. The following are the detailed answers (Responses can be reviewed in the attached file)

  1. Line 31: “…nanoparticles mass fraction of 20%.” – isn't this percentage too much for industrial use? Do such quantities make sense, and will the costs not be crazy?

Response: We thank the reviewer for her/his comment. The evaluation at lower percentages (0.01%, 0.1%, 1%, 5%, 10% and 20%) was presented in a previous published paper [1]. In that work, we evaluated different amounts of nanoparticles and found the lowest possible quantity for a possible future industrial scalation. In that evaluation, an impregnation at mass fraction of 1% (or higher value) allows an improvement in the materials performance for CO2 adsorption. The sandstone has an almost null adsorption capacity, therefore, the improvement of the adsorption capacity of the medium depends  on the nanoparticles amount (and their chemical-physical properties). For this reason, 10% or 20% have a better impact over adsorption capacity, but this percentage could be less based on our results previous [1].

This work does not cover industrial escalation or economic evaluation, but we recognize that the scaling considerations are relevant and will be considered for future articles based on the single well or reservoir scale simulations. However, in this work we considered other important variables for possible industrial application such as 1) Nanofluid composition (nanoparticles and water as carrying fluid) and 2) A simple impregnation method by means of a soaking process, which could be scaled at the industrial level [2].

On the other hand, the conventional CCS has technical and economical limitations, where the CO2 separation (purification) stage can represent a 70-80% of project cost. In conventional CCS, this stage is carried out by means of absorption or adsorption process [3-5], where the sorbent amount is higher than adsorbate and requires regeneration, which also, could affect the operation cost and the material lifetime.

Also, the materials presented in this work are synthesized from by-product (biomass), which could decrease the reagent cost. Although the synthesis method is carried out hydrothermally, the material can be considered as a green material and also this has a natural selectivity and competitive adsorption capacity, which could make it stand out among other materials synthesized by other cheaper methods (In previous paper, we evaluated a sol-gel method [1]). So, the carbon nanomaterials were selected by their good performance, and it could be economical and technically viable.

[1] E. Rodriguez Acevedo, F. B. Cortés, C. A. Franco, F. Carrasco-Marín, A. F. Pérez-Cadenas, V. Fierro, et al., "An Enhanced Carbon Capture and Storage Process (e-CCS) Applied to Shallow Reservoirs Using Nanofluids Based on Nitrogen-Rich Carbon Nanospheres," Materials, vol. 12, p. 2088, 2019.

[2] Franco-Aguirre, M., et al., Interaction of anionic surfactant-nanoparticles for the gas-wettability alteration of sandstone in tight gas-condensate reservoirs. Journal of Natural Gas Science and Engineering, 2018. 51: p. 53-64.

[3] Balat, H. and C. Öz, Technical and Economic Aspects of Carbon Capture a Storage-A Review. Energy Exploration & Exploitation, 2007. 25(5): p. 357-392.

[4] Gough, C., State of the art in carbon dioxide capture and storage in the UK: An experts’ review. International Journal of Greenhouse Gas Control, 2008. 2(1): p. 155-168.

[5] Gough, C., Carbon capture and its storage: an integrated assessment. 2016: Routledge.

  1. Keywords: - I think it's too much?

Response: We thank the reviewer for her/his comment made. The keywords are revised and following the comment, the “carbon nanospheres” has been eliminated. In this moment, the paper has five words (Ten keywords are allowed by the Journal’s instructions).

  1. Line 37: isn't it worth adding even a note about this global warming? Is it real global warming or maybe another period (in Earth's life) of rising and falling temperatures? As the authors believe - maybe it's worth mentioning?

Response: We thank the reviewer for her/his suggestion. The beginning of the introduction section was modified to expand the information about global warming. In this case, some important and recent references were considered. The new paragraph (lines 40-47) is:

“The global warming is still controversial topic worldwide, where its origin is mainly attributed to 1) Planet Earth has natural cycles that cause changes in global temperature, and 2) Human activities (anthropogenic global warming), are causing the nature deterioration and a greenhouse effect. This has allowed a better compression of both research lines and how they can be connected. Anthropogenic global warming has gained importance in recent years because of scientific research and reports that show the relationship between the increase in gas emissions and the increase in global temperature [1-5]. Whereby, there are different greenhouse gases, but global warming is mainly related to large amounts of carbon dioxide emissions [6-11]”

 The new references were included, such as:

 [1] I. Medhaug, M. B. Stolpe, E. M. Fischer, and R. Knutti, "Reconciling controversies about the ‘global warming hiatus’," Nature, vol. 545, pp. 41-47, 2017.

[2] N. Saklani and A. Khurana, "Global Warming: Effect on Living Organisms, Causes and its Solutions," International Journal of Engineering and Management Research (IJEMR), vol. 9, pp. 24-26, 2019.

[3] W. R. Cline, "The economics of global warming," Institute for International Economics, Washington, DC, p. 399, 1992.

[4] Y. Kosaka and S.-P. Xie, "Recent global-warming hiatus tied to equatorial Pacific surface cooling," Nature, vol. 501, pp. 403-407, 2013.

[5] K. E. Trenberth and J. T. Fasullo, "An apparent hiatus in global warming?," Earth's Future, vol. 1, pp. 19-32, 2013.

  1. Line 50: Will this CO2injection not create various dangers in the future? Maybe the authors could refer to it in a short sentence?

Response: We thank the reviewer for the comment made. A new sentence was included (lines 67-70):

There are many geological media that could be used for e-CCS process. Nature geological media as gas/oil reservoir could be perfect to CO2 storage due to physical conditions, such as nature seals, which could avoid possible operation problems in the future” 

  1. Line 61: - authors should add information on how these materials do it?

Response: We thank the reviewer for the comment made. The phrase is changed in the lines 73-76:

On the other hand, the spherical structure and nanometric size These nanomaterials allow the conservation of the porous structure of the geological media, avoiding clogging of pores, which prevents operational problems due to the formation damage and the associated loss of injectivity [23, 24]”

  1. Line 70: - or just Colombian?

Response: We thank the reviewer for the comment made. The agriculture is important for many countries. The phrase is changed in the line 83:

In this sense, agriculture is an integral part for economy of many countries, in Colombia, the agriculture activities are developed in many rural areas that have been vulnerable to violent conflict [31, 32]”

  1. Table 2 - the quality of the presentation of the designs leaves much to be desired - the authors must improve it.

Response: We thank the reviewer for the comment made. The table was revised. The equations were revised, and the format was unified.

  1. Fig 2 and 3 - note that the authors format the drawings differently. This must be corrected - once, for example, there is bold lines and text, other times not.

Response: We thank the reviewer for the comment made. The figures are revised, their format was unified (Figures 2, 3, 5 and 8).

  1. Fig 3 - the spectra are of poor quality. You can mark the bands the authors write about (in fact correctly) in the drawing - because that's how the potential reader must guess where it is.

Response: We thank the reviewer for the comment made. The figure was improved, and the main bands were marked.

  1. You can slightly expand the summary section - so that it more reflects the work because it has little to do with it.

Response: We thank the reviewer for her/his comment. We have a limit for abstract section of 200 words. Whereby, the abstract has been modifying to improve the section, such as has been proposed by reviewer. The new version is:

It is possible to take advantage of shallow reservoirs (< 300 m) for the CO2 capture and storage in post-combustion process by increasing the CO2 selective adsorption capacity. This process is called enhanced carbon capture and storage (e-CCS). In this process, it is necessary a nano-modifying agent to improve the chemical-physical properties of geological media, which allows increasing the performance of CO2 selective adsorption. Hence, this study presents the development and evaluation of carbon spherical molecular nano-sieves (CSMNS) from cane molasses for e-CSS. This is the first report in the scientific literature of CSMNS, due to its size and structure. The materials were synthesized by means of hydrothermal method and posteriorly characterized by several techniques for determining the size, structure and composition. Sandstone is used as geological media, and it is functionalized using a nanofluid, which is composed of CNMNS dispersed in deionized water, which were used for sandstone functionalization. Finally, CO2 or N2 streams were used for evaluating the adsorption process at different conditions of pressure and temperature. As the main result, the nanomaterial allows a natural selectivity to CO2, and the sandstone enhances the adsorption capacity by an incremental factor of 730 at reservoir conditions (50 °C and 2.5 MPa) using a nanoparticles mass fraction of 20%. This nanofluids applied to a novel concept of CCS for shallow reservoir open a novel landscape to control of industrial CO2 emissions.

Reviewer 2 Report

This work is interesting and potentially publishable in Nanomaterials. However, the following minor and major revisions need to be taken into account:

  1. The work is poorly written and needs to be improved. They were many English mistakes, the most of which have been corrected through the whole manuscript. However, there is need of better improvement.
  2. In introduction, a broader research has to be included in the literature, placing the state-of-art in the use of carbon-based nanomaterials and nanocomposite materials for enhanced carbon capture and storage  (e-CCS). A discussion of the use of other materials (no carbon-based) for e-CCS is necessary.
  3. In line 78 “but they were not selective for carbon dioxide as they showedhigha adsorption capacity” should change to “but they were not selective for carbon dioxide as they showed a high adsorption capacity”.
  4. In line 83 “for economy of many countries, in Colombia, the agriculture activities are developed” should change to “for economy of many countries. In Colombia, the agriculture activities are developed”.
  5. In line 96 “Therefore, the main objective of this manuscript is to synthesize” should change to “Therefore, the main objective of this paper is to synthesize”.
  6. In lines 103-106 “The material could be considered a molecular sieve due to it only allows the entry of CO2 molecules within its porous structure, which is evident in the high adsorption capacity, but does not allow the entry of N2 molecules, for which there is a poor adsorption capacity.” should change to “The material could be considered as a molecular sieve, due to the fact that, it only allows the entry of CO2 molecules within its porous structure, which is evident of high adsorption capacity, while it does not allow the entry of N2 molecules, thus exhibiting a poor adsorption capacity.”.
  7. In line 121 “and fiber” should change to “and fibers”. The same correction should be done in the table 1, ”Fiber” to be changed to “Fibers”.
  8. In line 132 “This material was selected as the template due to latex is composed of carbon molecules” should change to “This material was selected as the template considering that latex is composed of carbon-based molecules”.
  9. In lines 146-148 “was 1800:1, and the latex/carbon precursor mass ratio is 1:10. The latex nanoparticles are put in water with a 0.05 ml of Span 80 to disperse hydrophobic particles in the aqueous medium, at 25 °C for 4 h and 200 rpm. After that, the carbon precursor is added to the system and stirring for 30 min.” should change to “was 1800:1 and the latex:carbon precursor mass ratio was 1:10. The latex nanoparticles are placed in water with a 0.05 mL of Span 80 to disperse hydrophobic particles in the aqueous medium, at 25 °C for 4 h and 200 rpm. After that, the carbon precursor is added to the system followed by stirring for 30 min.”.
  10. In lines 167-169 “5556:1) is stirring at 200 rpm, 25 °C for 18 hours. Parallel, the latex nanoparticles are put in water with a 0.05 ml of Span 80 to disperse hydrophobic particles in the aqueous medium, at 25 °C for 4 h and 200 rpm. After that, the carbon precursor is added to the system and stirring for 30 min.” should change to “5556:1) is stirred at 200 rpm, 25 °C for 18 hours. Parallel, the latex nanoparticles are placed in water with a 0.05 mL of Span 80 to disperse hydrophobic particles in the aqueous medium, at 25 °C for 4 h 168 and 200 rpm. After that, the carbon precursor is added to the system followed by stirring for 30 min.”.
  11. In line 178 “[24], the minimum percentages to enhance the CO2 adsorption capacity are 10 and 20 %.” should change to “[24], the minimum nanoparticles percentages needed to enhance the CO2 adsorption capacity are 10 and 20 %.”.
  12. In line 233 the authors write “Initially, the material was cleaned by vacuum to remove gases and humidity.”. However, they don’t mention anything about the degassing protocol of the sample prior to measurements, by meaning the degassing temperature and the time.
  13. In line 242 “The model are presents in Table 2” should change to “The models are presented in Table 2”.
  14. In line 253 “This sections is divided in main three sections such as size and morphology,” should change to “This section is divided in three main parts such as size and morphology,”.
  15. In line 264 “presents a small size than latex template” should change to “presents a smaller size than latex template”.
  16. In line 268 “The latex template is presented in Figure 1-c. It can be observed nanospheres of size homogeneously distributed” should change to “The latex template is presented in Figure 1-c, where nanospheres of homogeneously distributed size can be observed,”.
  17. In line 264 “To have a comparison point” should change to “To have a point of comparison”.
  18. In lines 281-284 “Figure 1. SEM and TEM images at 5 kV of carbon spheres synthesized from cane molasses (CM) at different water:CM ratios of (a) 1800:1 (CNRON1) and (b) 3600:1 (CNRON2). (c) Latex spheres (Template). SEM and TEM images of carbon hollow nanospheres-CN.POL (water/resorcinol ratio of 5556:1; synthesized with latex template) (d) Image by SEM (e) Image by TEM” should change to “Figure 1. SEM images at 5 kV of carbon spheres synthesized from cane molasses (CM) at different water:CM ratios of (a) 1800:1 (CNRON1) and (b) 3600:1 (CNRON2). SEM image of Latex spheres (Template) (c). SEM (d) and TEM (e) images of carbon hollow nanospheres-CN.POL (water/resorcinol ratio of 5556:1; synthesized with latex template)”. The authors here have mixed the SEM with TEM images, the only TEM image existing in this manuscript is the last one, the 1(e)!!!
  19. I think that the manuscript will be further improved if the authors include at least a TEM image of the CNRON2, which is actually the material that the authors claim, from the experimental data, that is the carbon spherical molecular nano-sieve. The SEM image of this material (Figure 1(b) shows that we have to deal with a nanomaterial, but the TEM analysis would help having a better view regarding its morphology and if it has a hollow-core or not, etc.
  20. In line 291 “The aggregation is higher at pH values between 4.7 y 6.8” should change to “The aggregation is higher at pH values between 4.7 and 6.8”.
  21. In line 295 “Therefore an analysis de throat size distribution”, here it does not make any sense, should be changed to “Therefore an analysis of pore size distribution”???
  22. In line 299 “between 5 and 7) tends to add.” should change to “between 5 and 7) tends to increase.”.
  23. In lines 302-304 “for the detection. The hydrodynamic diameter was less for CN.RON2, where the particle size could be less than 100 nm because the aqueous system is, and here it hydrates and interacts with other particles.” should change to “for detection. The hydrodynamic diameter was less for CN.RON2, as it was expected, where the particle size could be less than 100 nm, but in aqueous system the particles are hydrated and interact with other particles, thus increasing a bit in size.”.
  24. In lines 315-316 “Around of 3600 cm–1 depicted the contribution of the O-H from ambient water. This band shown the stretching vibrations” should change to “Around 3600 cm–1 depicted the contribution of the O-H from ambient water. This band shows the stretching vibrations”.
  25. In line 323 “nitrogen surface groups, which is desirable to CO2 adsorption” should change to “nitrogen surface functional groups, which are desirable to CO2 adsorption”.
  26. Above the line 325 there are the two FTIR spectra of the materials CN.POL and CN.RON2, respectively. However, the y axis has to be renamed, it is wrong. The spectra are shown with their absorbance and not their %transmittance. Either you rename the axis (to Absorbance) and you leave the spectra as they are, or you don’t change the title but you have to correct the spectra (make them both spectra upside-down).
  27. In line 328 the authors write “The elemental analysis (CHON) was carried out for CN.RON2 and CN.POL (Table 4).”, However, they don’t mentioned which analyzer they have used for their measurements. They should change the phrase as “The elemental analysis (CHON) was carried out using the XXX analyzer for both CN.RON2 and CN.POL (Table 4).”.
  28. In line 330 “Taking account, the material comes biomass and has not undergone additional processes” should change to “Taking account that the material comes from biomass and has not undergone any additional treatment”.
  29. In lines 353-354 “which is agree with the reported by Mohamed et al. [60], Buonomenna et al. [61], and 353 Foley et al. [62].” should change to “which is in agreement with the reported by Mohamed et al. [60], Buonomenna et al. [61], and 353 Foley et al. [62] studies.”.
  30. In line 358 “is not larger than another material although the its high” should change to “is not larger than the other material (CN.POL) although its high”.
  31. In lines 368-369 “Based on the above, mainly to the obtained in the N2 and CO2 adsorption at -196 °C and 0 °C, CN.RON2 could have molecular sieve characteristics due to the null nitrogen adsorption.” should change to “Based on the above obtained data of the N2 and CO2 adsorptions at -196 °C and 0 °C, respectively, CN.RON2 could have molecular sieve characteristics due to its null nitrogen adsorption.”.
  32. In line 391 “Meanwhile, the nitrogen adsorption is null regard to the CO2 adsorption” should change to “Meanwhile, the nitrogen adsorption is null regarding to the CO2 adsorption”.
  33. In lines 395-396 “The nanosieves behavior is due to CN.RON2 only allows CO2 molecules to enter its porous structure” should change to “The nanosieve’s behavior of CN.RON2 can be attributed to its absorbing behavior, allowing only CO2 molecules selectively to enter in its porous structure”.
  34. In line 406 “Hence, the CN.POL was discarded for sandstone functionalization” should change to “Hence, the CN.POL was discarded for sandstone modification”. Here I think that the correct term is modification and not functionalization, because there is not any reaction between the SS and the corresponding Carbon spheres, it is more like a decoration of the SS with the carbon spheres.
  35. In line 419 “3.2.1. Adsorption onto sandstone before and after functionalization with CN.RON2 nanoparticles” should change to “3.2.1. CO2 adsorption onto sandstone before and after decoration (or modification) with CN.RON2 nanoparticles”.
  36. In lines 429-430 the authors write “The TGA analysis for SS-10 and SS-20 presents a variation of impregnation percentage. The real impregnation percentages are 8.6 and 20.8 %, respectively.”, but they don’t provide any TGA thermodiagram. I think that the authors should add the thermodiagrams in the supplementary material, thus changing the phrase to “The TGA analysis for SS-10 and SS-20 presents a variation of impregnation percentage. The real impregnation percentages are 8.6 and 20.8 %, respectively (see supplementary material, sFigure 1).”.
  37. In lines 431-433 “the sandstone composition is silica, which has an acid interaction type. For this reason, the interaction with CO2 is affected due to CO2 molecule also exhibits acid interaction type” should change to “the sandstone composition is mainly silica, which interacts with other molecules like an acid. For this reason, the interaction with CO2 is affected, considering the fact that CO2 molecule interacts also like an acid with other molecules”.
  38. In line 436 “The sandstone was evaluated in a preview work” should change to “The sandstone was evaluated in a previous work”.
  39. In lines 463-464 “To account for the synergistic effect of the proposed functionalization, the amount the theoretical amount adsorbed was calculated” should change to “To account for the synergistic effect of the proposed modification, the theoretical amount adsorbed was calculated”.
  40. In lines 467-468 “The nanoparticle comes from a biological precursor” should change to “The nanoparticle comes from a biomass by-product precursor”.
  41. In line 475 “The carbonization of polymer precursors usually prepares carbon molecular sieves membranes” should change to “The carbonization of polymer precursors usually prepares carbon-based molecular sieves membranes”.
  42. In line 478 “about the formation of membranes in this type of system” should change to “about the formation of membranes in this type of systems”.
  43. In line 482 “The reported structures are diverse, but until our knowledge there are no reports” should change to “The reported structures are diverse, but to the best of our knowledge, there are no reports so far”.
  44. In line 484 “the CO2 adsorption capacities reported in this chapter are competitive” should change to “the CO2 adsorption capacities reported in the present work are competitive”
  45. Finally, a quantitative comparison with previous results from similar systems should be included by adding a table. The table would be helpful, summarizing the main findings in literature in comparison with the achieved results.

Author Response

Dear Reviewer,

We thank the reviewer for her/his effort made for securing a prompt review of our manuscript and the opinion of publication after correction. Also, we thank for all revisions, all of them were included in the manuscript, and the text was improved. Additionally, our work has been improved in the English language. The attached document presents the detailed answers. This certainly encourages us to keep the good work.

Kind regards,

The authors

Reviewer 3 Report

In this paper the authors present the preparation and characterization of novel nanomaterials for enhanced carbon capture and storage process. The preparation method is based on the functionalization of standstone with carbon spherical molecular nano-sieves synthesized from cane molasses. Although this manuscript could be considered as an extension of previous works, where the authors have used carbon nanospheres as modifying agents of some reservoirs to improve the CO2 adsorption capacity, the paper have relevant results. The present study reveals that the carbon nanospheres obtained behave like molecular sieves, which allows selectivity to CO2 in mixtures of CO2/N2 under different pressure and temperature conditions. I consider that the work is properly carried out and the content would be of interest for the readers of Nanomaterials. I recommend acceptance of this paper. I also recommend the authors to revise:

  • Some spelling mistakes such as: "the materials was" (line 24), "tecniques" (line 25), "an highly" (line 480)
  • Figure numbering: Figure 6 appears twice
  • References: Subcripts have been omitted from the titles of some references (such as references 26, 29, 35, 49); Reference 2: 2019 appears twice;  Reference 25: Authors must be revised

Author Response

Responses to the Academic Editor suggestions for the manuscript: Carbon Spherical Molecular Nano-sieves from Biomass Residues applied to Enhanced Carbon Capture and Storage process (e-CCS) for Flue Gas Streams in Shallow Reservoirs

 Response: We thank the reviewer for her/his effort made for securing a prompt review of our manuscript and the opinion of publication after correction. Also, we thank for all revisions, all of them were included in the manuscript, and the text was improved. Additionally, our work has been improved in the English language. The following are the detailed answers.This certainly encourages us to keep de good work (The responses can be reviewed in the attached file).

  1. Some spelling mistakes such as: "the materials was" (line 24), "tecniques" (line 25), "an highly" (line 480)

Response: We thank the reviewer for the comment made. The mistakes in line 24 and line 25 were eliminated. It was even modified taking into account the observations of the other reviewer.

It is possible to take advantage of shallow reservoirs (< 300 m) for the CO2 capture and storage in post-combustion process by increasing the CO2 selective adsorption capacity. This process is called enhanced carbon capture and storage (e-CCS). In this process, it is necessary a nano-modifying agent to improve the chemical-physical properties of geological media, which allows increasing the performance of CO2 selective adsorption. Hence, this study presents the development and evaluation of carbon spherical molecular nano-sieves (CSMNS) from cane molasses for e-CSS. This is the first report in the scientific literature of CSMNS, due to its size and structure. The materials were synthesized by means of hydrothermal method and posteriorly characterized by several techniques for determining the size, structure and composition. Sandstone is used as geological media, and it is functionalized using a nanofluid, which is composed of CNMNS dispersed in deionized water, which were used for sandstone functionalization. Finally, CO2 or N2 streams were used for evaluating the adsorption process at different conditions of pressure and temperature. As the main result, the nanomaterial allows a natural selectivity to CO2, and the sandstone enhances the adsorption capacity by an incremental factor of 730 at reservoir conditions (50 °C and 2.5 MPa) using a nanoparticles mass fraction of 20%. This nanofluids applied to a novel concept of CCS for shallow reservoir open a novel landscape to control of industrial CO2 emissions.”

In line 480, the phrase was corrected it a highly promising material for the e-CCS process” (New line 498).

  1. Figure numbering: Figure 6 appears twice

Response: We thank the reviewer for the comment made and we apologize for the typo. The figures were revised. The format was unified in Figures 2, 3, 5 and 8. Also, The Second Figure 6 was renumbered, whence the lines 437-440 and 454-455 changed:

The CO2 adsorption over the sandstone impregnated with CN.RON2 nanoparticles at mass fractions of 10% and 20 % (CN.RON2), and 25 °C and 50 °C is shown in Figure 7. The CO2 adsorption capacity is considerably higher than the adsorption capacity of the sandstone without surface modification.”

  1. References: Subscripts have been omitted from the titles of some references (such as references 26, 29, 35, 49); Reference 2: 2019 appears twice; Reference 25: Authors must be revised

Response: We thank the reviewer for her/his comment. The references were revised and modified.

Round 2

Reviewer 2 Report

The authors have improved the new version of the manuscript according to my instructions, and therefore I am pleased to inform them that I will accept the manuscript for publication in its current form. Just if you can change the following before publication (the numbering of the line concerns to the final version, v3, of the manuscript):

In line 363 “which is agreement with the reported by Mohamed et al.” should change to “which is in agreement with the reported by Mohamed et al.”.

Just add the "in".